# Polyploid Cancer Cell Models in Drosophila

**DOI:** 10.3390/genes15010096

**Published:** 2024-01-14

**Authors:** Yuqing Wang, Yoichiro Tamori

**Affiliations:** Department of Molecular Oncology, Kyoto University Graduate School of Medicine, Kyoto 606-8501, Japan

**Keywords:** polyploidy, polyploid cancer cells, tumorigenesis, stress resistance, *Drosophila*

## Abstract

Cells with an abnormal number of chromosomes have been found in more than 90% of solid tumors, and among these, polyploidy accounts for about 40%. Polyploidized cells most often have duplicate centrosomes as well as genomes, and thus their mitosis tends to promote merotelic spindle attachments and chromosomal instability, which produces a variety of aneuploid daughter cells. Polyploid cells have been found highly resistant to various stress and anticancer therapies, such as radiation and mitogenic inhibitors. In other words, common cancer therapies kill proliferative diploid cells, which make up the majority of cancer tissues, while polyploid cells, which lurk in smaller numbers, may survive. The surviving polyploid cells, prompted by acute environmental changes, begin to mitose with chromosomal instability, leading to an explosion of genetic heterogeneity and a concomitant cell competition and adaptive evolution. The result is a recurrence of the cancer during which the tenacious cells that survived treatment express malignant traits. Although the presence of polyploid cells in cancer tissues has been observed for more than 150 years, the function and exact role of these cells in cancer progression has remained elusive. For this reason, there is currently no effective therapeutic treatment directed against polyploid cells. This is due in part to the lack of suitable experimental models, but recently several models have become available to study polyploid cells in vivo. We propose that the experimental models in *Drosophila*, for which genetic techniques are highly developed, could be very useful in deciphering mechanisms of polyploidy and its role in cancer progression.

## 1. Introduction

Polyploid cells are frequently observed in advanced cancer, particularly after standard cancer treatment such as anticancer drugs and radiation therapy [1]. This suggests that polyploid cells lurking in a cancer tissue possess a superior ability to withstand environmental stress, making them more likely to survive anticancer therapies. The polyploid cells in cancer tissues, commonly termed polyploid giant cancer cells (PGCCs), but also referred to as blastomere-like cancer cells, osteoclast-like cancer cells, pleomorphic cancer cells, large cancer stem cells, and polyaneuploid cancer cells (PACCs), are thought to play an important role in tumor progression [2]. Recent studies have increasingly revealed that polyploid cells emerge under stress conditions such as damaged tissue repair or regeneration and aging [3,4,5,6,7,8,9,10,11], and thus polyploid cells can be found not only in tumors but in various normal tissues. Polyploid cancer cells likely originate from those polyploid cells that emerged in a tissue as a result of stress conditions [12,13]. At the same time, many studies on polyploid cancer cells have reported that polyploidy is frequently induced after radiation treatment [14]. However, it is still unclear why polyploid cancer cells are found in so many malignant tumors and what makes polyploid cells so tenacious. For mammalian cell culture systems, purifying and culturing polyploid cells in vitro is challenging, as they exist in tumor tissues for a short period after cancer treatment, and, currently, there are no specific markers available to isolate them from a mixture of numerous diploid cells and a few polyploid cells. Moreover, even if polyploid cells can be successfully isolated from a tissue, they gradually reduce their ploidy and produce aneuploid offspring through mitotic cell division (depolyploidization) during in vitro culture [15,16], further complicating their study. Therefore, it is crucial to explore new experimental models that can effectively represent polyploid cells to elucidate their characteristics and roles in cancer progression, therapeutic resistance, and other related biological processes. In this review, we present the experimental systems in *Drosophila melanogaster* as models for the investigation of polyploid cancer cells and provide an overview of the latest research progress made in understanding polyploid cells using these models.

## 2. Polyploid Cells

Polyploid cells are defined as a cell containing more than two sets of homologous chromosomes, which shows a larger nucleus and cell volume compared with a normal diploid cell. Most multicellular organisms including humans employ sexual reproduction in which the fertilization of two different haploid gametes produced through meiotic cell division generates diploid zygotes. The diploid zygotes (fertilized eggs) undergo numerous mitotic cell division which produces a copied daughter cell to form a multicellular adult body. With this system, most of the cells in our body keep diploidy, and the diploidy is maintained from generation to generation. At the same time, genetically programmed systems employ polyploidy in development to achieve particular purposes such as quick organ growth during larval stages, production of a large amounts of specific proteins, or cellular behaviors that consume a lot of energy [17,18]. Recent studies also have shown that polyploid cells emerge in stress situations such as damage-induced tissue repair in postmitotic tissues known as “compensatory cellular hypertrophy” [3] and “wound-induced polyploidy” [4]. These postmitotic tissue-repair systems were found in *Drosophila* and similar mechanisms have been observed in various vertebrate organs as well [6,8,9,10,11]. Furthermore, it has been shown that the number of polyploid cells gradually increases with aging [5].

Observations of polyploid cells or giant cells in cancer tissues have been reported from over 150 years ago [1,19] and have intrigued cancer biologists ever since [1,20,21]. Such polyploid cancer cells have been shown to have the capability of proliferation, mitotic cell division, and genetic diversification and thus are considered to play multiple functions as a stem-like cell in tumorigenesis [22], tumor malignancy [23], and therapeutic resistance [2,24,25].

### 2.1. Polyploid Cells in Cancer

Numerous studies have highlighted the occurrence of polyploidy in various types of human cancer, including breast cancer [23,26], nasopharyngeal carcinoma [27], melanoma [22], lung cancer [28,29,30], rectal cancer [25], colorectal cancer [31], and glioblastoma multiforme (GBM) [24]. Importantly, the presence of polyploid cells in these cancers is associated with a poorer prognosis [32].

Polyploid cells are not exclusive to tumors but can also be observed in healthy human tissues. For instance, megakaryocytes increase their ploidy by a process of endomitosis [33,34], extraembryonic trophoblast cells undergo several rounds of endoreplication to become polyploid [35,36], and adult human livers consist of approximately 40% polyploid cells originated through incomplete cytokinesis [37,38,39]. A recent study suggests that liver polyploidy limits the development of hepatocellular carcinoma (HCC), a common type of liver cancer. In this context, the presence of healthy polyploid cells plays an active role in safeguarding against cancer [40,41]. Polyploid hepatocytes provide a strong buffer against loss or mutation of tumor suppressor genes, which may be due to a sufficient tumor suppressor gene reserve from the multiple genome copies [42]. At the same time, a recent study using in vivo lineage tracing in mouse livers has shown that polyploid hepatocytes readily initiate tumorigenesis through multiple rounds of cell division and that the ploidy reduction frequently occurs during an early phase of tumorigenesis [43].

Consequently, it becomes more important to differentiate the functions of polyploid cells in cancer and those in healthy tissues. The time point (before or after polyploid cell formation) of key gene loss or mutation may be an important factor in determining whether polyploid cells have a positive or negative effect on tumorigenesis and tumor malignancy.

### 2.2. Types of Polyploid Cells

Based on the morphological phenotypes of polyploid cells, we can divide them into two types: single-nuclear cells and bi-nuclear cells. Even if the cellular phenotype of a polyploid cell is of the same type, the process of polyploidization could be different. According to the mechanism of polyploidization, we can classify them into two categories: intracellular duplication and intercellular merge. The intracellular duplication can be further divided into four subclasses: endocycle, mitotic slippage, endomitosis, and cytokinesis failure. The intercellular merge can be further divided into two subclasses: cell fusion and cell cannibalism (cell engulfment or entosis) [44,45,46] (Figure 1).

One of the most common types of polyploidization is endocycle (endoreplication or endoreduplication cycle)-induction of single-nucleus polyploid cells in which cells exit their mitotic cell cycle in response to developmentally programmed or environmental signals and undergo cell cycle transition from mitosis to endoreplication [47]. In the endocycle, the mitotic cell cycle skips the M phase, therefore, the cells do not undergo mitotic cell division but duplicate their genome. Among many regulatory genes taking part in this process, Cdh1, *fizzy-related* (*fzr*) in *Drosophila*, plays a key role in the M-phase skipping [48]. Mechanically, Cdh1/Fzr combines with the anaphase-promoting complex/cyclosome (APC/C) complex and promotes the ubiquitination-mediated degradation of Cyclin B1 leading to the M-phase skipping [47,49,50]. This endocycle-mediated polyploidization is endogenously employed in various organs in *Drosophila* [51]. It has also been shown that in *Drosophila*, overexpression of Fzr in mitotic cells proliferating in developing tissues such as larval imaginal discs induces mitotic-to-endocycle transition and polyploidization [48,52]. The endoreplication-mediated polyploidization is prevalent in many different cancer types [49,53].

Another common type of polyploidization is cytokinesis failure-induced bi-nuclear cell which prematurely finishes anaphase and forms two nuclei but fails in cytokinesis [54]. Some types of cancer cells such as melanoma often show this type of polyploidization through a deactivation of RhoA, one of the critical cytokinesis regulators [22]. Similar polyploidization can be observed in some physiological situations. For example, epithelial cells in *Drosophila* male accessory glands undergo mitosis without cytokinesis. This is followed by an additional round of DNA replication of each nucleus resulting in the formation of octaploid cells with two tetraploid nuclei (4C + 4C) [55]. Although this binucleation process in the accessory gland cells is similar to cytokinesis failure, this is an endogenously programmed system but not a failure.

### 2.3. Causes and Consequences of Polyploidy in Cancer

In the context of cancer development, polyploidization processes could be triggered by different factors such as internal genetic mutations or external survival pressure. In the case of internal genetic mutations, a specific gene mutation can induce polyploid cell formation. One such mutation commonly found in melanoma is BRAF^V600E^, which is involved in the generation of polyploid cells and the promotion of tumorigenesis [22]. The underlying mechanism of this process involves BRAF^V600E^ triggering the formation of excess centrioles and centrosomes, leading to the activation of a Rac-family small GTPase Rac1 (Ras-related C3 botulinum toxin substrate 1). As a result, the Rac1 downstream RhoA is inhibited, and cytokinesis fails to occur, contributing to the generation of polyploid cells [22]. In the case of myeloproliferative tumors, loss-of-function mutations in Enhancer of Zeste 2 (EZH2), the enzymatic component of polycomb repressive complex2 (PRC2), are involved in the suppression of cell proliferation, polyploidization, and platelet formation through regulation of CDKN1A gene expression during megakaryocyte maturation [56].

Apart from internal genetic mutations, external survival pressure can also induce polyploidization. Various stressors encountered during therapeutic cancer treatments, including systemic chemotherapy [57], radiation [14], and targeted therapies [27,58], have been observed to trigger the generation of polyploid cells. Notably, a significant proportion of tumor cells that survive treatment exhibit a polyploid phenotype [24,25]. We can also hypothesize that the polyploid cells were originally present in cancer tissues. If this is the case, these stressors may not be the cause of polyploidization but may promote the survival of polyploid cells in a situation where genetically diverse cell population including diploid, aneuploid and polyploid cancer cells compete with each other. The ability of polyploid cancer cells to generate immense genetic diversity through polyploid mitosis may explain the reason why they can increase the probability of survival of progeny cells (the mechanisms will be discussed in Section 4.3). In this sense, it would be important not to confuse the cause of the polyploid cell generation with the consequence of stress on polyploid cancer cells.

Regarding endogenously developed polyploid cells, a series of studies conducted on *Drosophila* ovarian follicle cells have demonstrated that the endocycling polyploid cells possess a robust capability to resist apoptotic response to DNA damage induced by γ-ray irradiation and that this radiation resistance of endocycling cells is supported by the low level of p53 and silencing of other pro-apoptotic genes [59,60,61,62]. Although the *Drosophila* ovarian follicle cells are non-cancerous polyploid cells, deciphering the molecular mechanisms of their radiation resistance might give us insight into the therapeutic resistance of polyploid cancer cells.

## 3. Advantages of the *Drosophila* Model in Polyploid Cell Investigation—Easy Access

The investigation of polyploid cells in mammalian systems has long been attracting researcher’s interest but remains an unsolved problem. Acquiring cultured polyploid cells in vitro poses several technical challenges. First, stable cultured cell lines of polyploid cancer cells are not readily available. Second, obtaining polyploid cancer cells from tumor tissues and culturing them in vitro is difficult due to their limited presence and low numbers within the tumor. Additionally, the lack of specific polyploid cell markers makes it challenging to isolate and purify these cells. Although drug or radiation treatments can induce polyploid cancer cells formation in vitro, the polyploid cancer cells only exist for a short period in a mixed environment of polyploid and diploid cells; subsequently, they undergo ploidy reduction through multiple divisions [63,64,65].

In vivo studies using mouse models are also challenging, as the common approach involves the generation of transgenic strains and breeding mice for genetic crosses which incur high costs. Furthermore, the long life cycle of mice adds to the inconvenience of time required for genetic experiments. An ideal research model for studying polyploidy would involve simple genetics to produce stable polyploid cells, cost-effectiveness, and time efficiency. For these reasons, *Drosophila* models appear to meet these criteria exceptionally well.

The current *Drosophila* model offers two types of polyploid cells: artificially induced and naturally developed polyploid cells. The subsequent section will delve into the details of these two approaches.

### 3.1. Polyploid Tumor Model—Larval Salivary Gland

The imaginal ring of *Drosophila* larval salivary glands is composed of adult progenitor cells and undergoes proliferation during the late larval stages to replace the neighboring larval tissues for adult organ formation [66]. It has been shown that the cells in the narrow transition zone that reside at the border between the posterior end of the imaginal ring and the secretary polytene salivary gland cells endogenously undergo endoreplication to become polyploid [67]. These polyploid cells in the transition zone have an endogenously high activation level of Janus kinase/signal transducer and activator of transcription (JAK/STAT) signaling, c-Jun N-terminal kinase (JNK) signaling, and JNK-induced matrix metalloproteinase 1 (MMP1) expression all of which are involved in oncogenesis. These tissue characteristics confer the cells in this area oncogenic potential, which is called “tumor hotspots” [68,69] (Figure 2).

Misexpression of an active form of Notch, the Notch intracellular domain (NICD), in salivary glands during the third instar larval stage induces tumorigenesis. In the tumorigenic salivary glands, a hyperplastic overgrowth is induced in the diploid imaginal ring cells, but neoplastic tumor phenotypes including multilayer growth and polarity loss are induced in the polyploid transition zone [67]. The re-entry to mitosis followed by continuous cell division of the polyploid transition-zone cells in response to the Notch signaling overactivation is key to tumor initiation and progression [70]. These tumor cells display a high degree of variations in ploidy levels and show a marked increase in chromosomal instabilities and aneuploidy as the tumors continue to grow. Interestingly, some genes involved in the DNA damage response (DDR) pathway are necessary for tumorigenesis, like *Meiotic recombination 11* (*MRE11*), *Replication protein A1* (*RPA1*), *Structural maintenance of chromosomes protein 5* (*SMC5*), and DDR kinase—*meiotic 41* (*mei-41*, the *Drosophila* homolog of ATR), which imply the important function of DNA repair factors in this tumorigenesis [70].

### 3.2. Polyploid Tumor Model—Wing Imaginal Disc

Imaginal discs are larval epithelial tissues developing into particular parts of adult insect bodies. Among several imaginal discs of *Drosophila* larvae, wing discs have been commonly used as an experimental model for organ growth, pattern formation, and tumorigenesis [71]. Recently, a genetic experimental study has shown that an overexpression of EGFR combined with microRNAs (*bantam* or *miR-8*) causes cancerous phenotypes in the *Drosophila* wing discs including epithelial polarity defect and invasive migration, which provides us with a polyploid tumor model in this epithelial tissue [72] (Figure 2). Specifically, the tumor is made of polyploid cancer cells. Although overexpression of EGFR alone causes benign tissue hyperplasia in the epithelial tissues, a combinatorial overexpression of *miR-8* and EGFR induces polyploid tumors. In this model, *miR-8* expression causes genome instability by downregulating the expression of the Septin family protein Peanut. Peanut is similar to mammalian Septin7, and its depletion induces binuclear cell formation through cytokinesis failure [73]. The cytokinesis failure combined with EGFR overactivation leads to the formation of neoplasia with polyploid cancer cells (Figure 2).

One disadvantage of using wing imaginal discs as a tumor model is the limited time available to observe cancer progression during larval stages (~7 days). Recently, however, a continuous culture system that can allow us to observe the tumor progression over a much longer period has been developed [74]. This experimental system is to perform generational allotransplantation of imaginal disc tumors into the abdomen of adult hosts. This technique will not only allow us to observe the transition of cancer tissues over time but will also be a very powerful tool for studying the relationship between cancer and its host body.

### 3.3. Artificially Induced Polyploid Cells—Wing Imaginal Disc

The artificially induced polyploid cell models in *Drosophila* are based on the Gal4-UAS binary expression system, which facilitates the expression of a specific cell-cycle regulatory gene such as *fizzy-related* (*fzr*, mammalian homolog of CDH1). In the Gal4-UAS system, Gal4 is a transcriptional activator that binds to the UAS enhancer sequences, thereby driving the expression of a gene of interest linked to UAS. Gal4 is typically fused to an enhancer sequence of a gene that is spatiotemporally active in specific organ regions, thereby controlling Gal4 expression in the specific region [75]. For instance, the *engrailed-Gal4* (*en-Gal4*) is associated with the *engrailed* (*en*, a homeodomain selector gene) enhancer sequence, which is specifically expressed in the posterior compartment of the organ, enabling the induction of a UAS-gene expression in the posterior side of wing imaginal discs [76] (Figure 3).

*fzr* is a regulatory gene involved in endoreplication, and its overexpression can induce the endocycle-mediated polyploidization [77,78]. Mechanistically, Fzr interacts with the APC/C (anaphase-promoting complex/cyclosome) and induces a transition from mitotic to endocycle by degrading mitotic cyclins and skipping the M phase [48,77,79]. We can employ the *en-Gal4* to induce the misexpression of *fzr* in the posterior compartment of the wing discs, resulting in the generation of polyploid cells. This approach allowed us to obtain diploid cells in the anterior side and polyploid cells in the posterior side of the same wing disc, facilitating a convenient comparison of phenotypes and molecular regulation simultaneously.

To control the duration of *fzr* expression, we can utilize a temperature-sensitive form of Gal80 (Gal80ts). Gal80 blocks Gal4-mediated transcriptional activation by binding to its transcriptional activation domain. The Gal80ts is functional to inhibit Gal4 activation at 18 °C but ceases to function at 29 °C [80]. In the experimental setting, the fly larvae bearing *en-Gal4*, *UAS-fzr*, and ubiquitous Gal80ts (tub-Gal80ts) are kept at 18 °C for 6 days while waiting the eggs to develop into first instar larvae. Then, we shift the temperature to 29 °C to start Gal4-mediated *fzr* expression and keep them at 29 °C for 2 days before the dissection of wing discs from third instar larvae (Figure 3).

Prolonged misexpression of *fzr* leads to developmental defects in the wing disc. For example, when we keep them at 29 °C for 3 days after the temperature shift, the growth of the posterior compartment is not enough to develop into its normal size probably because the polyploid cells show apoptosis and do not proliferate. As a result, diploid cells in the anterior compartment proliferate to compensate for the undergrown posterior side of the wing disc. This is an interesting phenotype because the mechanism by which long-time *fzr* causes polyploid cell apoptosis is unknown. It is quite likely that the apoptosis of those polyploid cells resulted from chromosomal instability and mitotic catastrophe.

There is one more thing to take into consideration when we use this experimental model. The ploidy of those polyploid cells induced in the posterior compartment of wing discs is not uniform. The nuclear size of polyploid cells in the posterior compartment 2 days after temperature shifting is larger in the wing pouch than in the proximal hinge regions. This difference in ploidy level between these regions might be attributable to the differential regulation of Cyclin E in this tissue [52].

Using this experimental model, we can also investigate the mitotic cell division of polyploid cells by canceling the misexpression of *fzr*. When we shift back the temperature from 29 °C to 18 °C, the Gal4-driven *fzr* expression ceases and the polyploid cells go back to the mitotic cycle. When polyploid cells undergo mitotic cell division, there is a high probability of chromosome missegregation that could produce aneuploid daughter cells. There are many apoptotic cells observed in the posterior compartment after canceling the *fzr* expression, probably because aneuploid cells produced through the chromosome missegregation cannot gain survival advantages. Therefore, this experimental model also provides a tool to study how polyploid cells undergo depolyploidization that produces stable diploid or unstable aneuploid daughter cells.

### 3.4. Endogenously Developed Polyploid Cells—Ovarian Follicular Epithelia

In *Drosophila*, there are many organs endogenously bearing polyploid cells. One such well-studied example is, as already mentioned, the ovarian follicular epithelium composed of the monolayer of epithelial cells covering germline cells in an egg chamber [81]. The follicle cells endogenously undergo the cell-cycle transition from mitotic to endoreplication cycle and generate polyploid cells during the mid-stage of the oogenesis (stages 7 to 10) [82]. This natural process allows us to observe both diploid and polyploid cells within the same ovariole [71]. The mitotic-to-endocycle transition is triggered by the Delta-Notch signaling pathway activated at stage 7, and Fzr plays a crucial role in this cell-cycle switch as a downstream event of Notch signaling activation [77] (Figure 2).

In this experimental model, we can use a pan-follicle cell driver *traffic-jam-Gal4* (*tj-Gal4*) that can induce UAS-gene expression in every follicle cell of egg chambers in all oogenesis stages [83,84]. Although the *tj-Gal4* is expressed in ovarian follicle cells, its expression can be observed also in several other organs during different larval stages [85]. Therefore, the combination of Gal80ts and Gal4-UAS system with the temperature shifting can be utilized to activate the *tj-Gal4* expression in the adult stage. To bypass the larval stages, the flies bearing *tj-Gal4* and UAS-gene should be maintained at 18 °C after fertilization to the adult stage. Two days before dissection, we transfer the adult female and male flies to the same fooded vial to stimulate ovary maturation of the female flies and keep them at 29 °C.

Unlike the imaginal-disc experiment, we do not need to misexpress an additional gene such as *fzr* to induce polyploidy in follicle cells because they endogenously undergo endocycle-mediated polyploidization. This makes the genetic cross scheme to have another genetic manipulation in the polyploid cells simpler and allows us to examine easily what genetic backgrounds affect the nature of polyploidy. At the same time, it is important to note that egg chamber development is so quick (the oogenesis completes within 3 days) that we need to be careful with precise control of Gal4 expression and that the spatiotemporally varying developmental signals according to oogenesis stages may affect the comparison between diploid and polyploid follicle cells. Nevertheless, the advantages of using endogenously programmed polyploidization system like ovarian follicle cells is that the ploidy is controlled (16C in the stage 10 egg chamber) and the number of polyploid cells remains relatively consistent within the same oogenesis stage (approximately 650 follicle cells in the stage 10 egg chamber) [82].

*Drosophila* ovarian follicular epithelia offer us yet another polyploid tumor model. Using *tj-Gal4* with the *Gal80ts* system to control the expression of *UAS-RNAi* for a neoplastic tumor suppressor gene *lethal giant larvae* (*lgl*), we can induce epithelial polarity defect and multilayer formation in the follicular epithelia during midoogenesis [71,86]. The basally located cells in the multilayered epithelium cannot stop proliferating and become tumorigenic because they lose contact with the inner germ-line cells which are the source of the Notch signaling ligand Delta and thus the blockade of Notch signaling prevents the mitotic-to-endocycle transition [71]. These tumorigenic cells in the multilayer show a distinguishable variability in their ploidy that is enhanced when Notch signaling activation is ectopically induced (via co-expression of NICD) in the *lgl*-knockdown cells [86,87]. At the same time, ectopic expression of Snail-family transcriptional factor Escargot promoted in the tumorigenic *lgl*-knockdown cells enhances ploidy heterogeneity, likely via depolyploidization [86].

## 4. Features of Polyploid Cells

### 4.1. The Protein Synthesis and Metabolism in Polyploid Cells

Protein synthesis is one of the cellular metabolic processes. Several studies have shown the relationship between ploidy, cell size, and cellular metabolic rate [88,89,90]. Studies of artificially induced polyploid animals such as frogs and fish also have shown that the whole-body metabolic rate is lower in both larvae and adults [91,92]. Recently, a study using *Xenopus* embryos and tadpoles with different ploidy showed that ploidy affects metabolism by altering the cell surface area to volume ratio when cell size correlates with genome size [89]. When an activity of the Na^+^/K^+^ ATPase was inhibited, the metabolic difference between diploids and triploids was abolished, suggesting that less energy is required to maintain plasma membrane potential in triploids due to the lower cell surface area to volume ratio. Also, the similar low metabolic phenotypes of polyploid cells were consistent with the data in yeast [88] and zebrafish [93]. It would be important to confirm whether the metabolic rate alteration contributes to the higher stress resistance of polyploid cells especially to radiation tolerance.

One plausible explanation for the radiation tolerance of polyploid cells might be that polyploid cells have additional copies of the genome. Multiple copies of the genome in polyploid cells should synthesize more proteins including DDR-related proteins, which will confer the high ability of stress resistance on polyploid cells. According to this hypothesis, we have examined the O-propargyl-puromycin (OPP) assay to verify the protein synthesis rate in *Drosophila* wing discs and found that the protein synthesis rate of polyploid cells is higher than diploid cells (the relative quantification is 115%). Conversely, in the ovarian follicle cells, we found that the protein synthesis rate of endocycling polyploid cells examined by the OPP assay is less than that in mitotic diploid cells (the relative quantification is 63%). Interestingly, we also confirmed that both types of polyploid cells (artificially induced polyploid cells in wing discs and endogenously developed polyploid follicle cells) show a high ability of resistant to γ irradiation. From these observations, we cannot conclude whether the protein synthesis rate contributes to the irradiation resistance of polyploid cells. In order to solve this problem, the whole genome translation status in polyploid cells should first be verified using, for instance, ATAC-seq (Assay for Transposase-Accessible Chromatin sequencing), RNA-seq, and quantitative proteome analysis.

### 4.2. Stress Resistance

Polyploid organisms show a higher stress resistance, which has been shown in many plants such as crops and fruit-bearing plants under exposure to adverse environmental conditions [94,95]. Although polyploid species are rare in animals, some amphibian species of sexually reproducing polyploid lineages represent a good example. The geographically different distribution of the Australian burrowing frog genus *Neobatrachus* comprised of diploid and polyploid species shows that the polyploid species are better adapted to drier regions than are diploid species, suggesting that the polyploid species could invade harsher ecological niches [96,97]. Polyploid plant species have also been shown to be more invasive than closely related diploid species [98]. Similar organismal stress tolerance can be observed in unicellular organisms [99,100,101]. The human liver is the main organ to deal with toxin, thus the hepatocytes are constantly exposed to genotoxins and thus need a strong capability of stress resistance. This detoxification function might be one of the reasons why mammalian livers contain a significant number of polyploid hepatocytes (approximately 40% of adult human and 90% of adult mouse hepatocytes) [39,102]. Also, the proportion of polyploidy increases with cellular stress, aging, and in hepatectomy-induced regeneration process, which could be the consequence of compensatory cellular hypertrophy [12].

The high level of stress tolerance has been verified also in the *Drosophila* polyploid models. For instance, *Drosophila* ovarian follicle cells in the endocycling stages (oogenesis stages 7–10) did not apoptose after γ-ray irradiation [61]. The genetic experiments with artificially induced polyploid cells in the wing imaginal disc model also showed a similar characteristic of polyploid cells (we discuss this topic in Section 5). Therefore, stress resistance is a common characteristic of polyploid cells. However, why polyploid cells have much higher stress tolerance and how polyploid cells resist stress so effectively have not been explained yet. Considering the large number of haploinsufficient genes, an increased copy number of each gene might be able to allow polyploid cells to survive stress.

### 4.3. Polyploid Mitosis and Chromosome Instability

After exposure to strong stress such as γ irradiation, polyploidization is induced through mitotic catastrophe, and subsequently, the polyploid cells undergo cell division and reduce ploidy which is called polyploid mitosis or depolyploidization [16,103,104]. Studies have shown that the process of polyploid mitosis is error-prone and thus it contributes to chromosomal instability and tumor progression [43,70]. In fact, depolyploidization sometimes confer a high invasion and stress-resistance capability [103,105,106]. It has been reported that the expression of genes involved in meiosis is upregulated in breast, cervical, and colon cancer cells after irradiation [65]. Similar meiosis-specific gene expression was observed in the polyploid tumor model of *Drosophila* salivary glands in which depolyploidization of polyploid tumor cells occurs spontaneously without irradiation [70].

Typically, polyploid cells have multiple centrosomes (more than 2, the number is dependent on their ploidy), because they duplicate not only their genome but also centrosomes during the S phase [107,108]. Centrosome retention is associated with future polyploid mitotic potential [13,108,109,110]. Polyploid cancer cells bearing multiple centrosomes start mitosis with multipolar spindles, but in general, multipolar spindles are not observed during anaphase due to processes called spindle assembly checkpoint (SAC) and centrosome clustering [109,110,111]. The SAC is a surveillance mechanism that monitors the attachment of chromosomes to the spindle microtubules to ensure accurate chromosome segregation in mitosis [112]. When kinetochores are unattached, they generate a checkpoint signal that delays the onset of anaphase. Once all kinetochores are attached, the checkpoint signal ceases, ensuring that the cell cycle proceeds to anaphase after all chromosomes are bound to mitotic spindles [113]. Therefore, the SAC should play an important role in the prometaphase of polyploid cells which start mitosis with an increased number of centrosomes and a risk for multipolar spindle formation.

Centrosome clustering is a process in which extra centrosomes are brought together to form two functional spindle poles, leading to the formation of a bipolar spindle [114,115,116]. This is another essential process for proper chromosome segregation; thus this process should help prevent the formation of multipolar spindles in polyploid mitosis. It has been shown that various human cancer cell lines harboring extra centrosomes undergo spindle bipolarization by centrosome clustering before anaphase onset, thereby avoiding multipolar spindles during anaphase [109]. The presence of multipolar spindles at the beginning of polyploid mitosis causes merotelic attachments which leads to a high frequency of chromosome instability. Notably, merotelically attached chromatids are pulled to opposite poles of the cell and become trapped in the central spindle, resulting in a lagging chromosome [110,117]. In the following cytokinesis, the lagging chromosomes are randomly segregated into a daughter cell, leading to the production of two aneuploid cells with high probability [118,119].

Therefore, polyploid mitosis could explosively enhance the genomic diversity of a cell population. At the same time, most aneuploid cells have lower survival advantages; many of them die or are eliminated through cell competition with neighboring normal cells. Even if most of the aneuploid cells die, a few aneuploid cells with survival advantages will successfully survive and generate offspring cells [16,120]. These aneuploid survivors should have higher stress resistance and thus be more competitive in the environment. This could be one of the main reasons why malignant tumor progression occurs frequently after treatment.

On the other hand, a series of studies using the rectal papillar cells in *Drosophila* larvae which naturally undergo extra chromosome duplications before mitosis have shown that these polyploid cells are programmed to avoid polyteny by setting the extra chromosome copies apart from each other before mitosis [78,121,122]. This particular type of chromosome separation with the distinct form of interphase cohesin regulation can eliminate its associated mitotic defects and ensures mitotic fidelity after genome reduplication [123]. This suggests that specific mechanisms to avoid mitotic catastrophe are employed in naturally developed polyploid cells which undergo programmed mitosis after polyploidization.

## 5. Study of Radiation Resistance of Polyploid Cells in *Drosophila* Models

Although flies and mammals have many differences in the appearance of organs and the details of signaling pathways, they share deep homology of essential proteins, tissue and cell structures, physiological systems, and tumor phenotypes [124]. The powerful genetic tools and the ease of handling in a laboratory make *Drosophila* one of the most useful model systems to study polyploid cells in vivo in the context of cancer therapy including radiation and chemical therapies. Although some targeted drugs or inhibitors cannot be applied to *Drosophila* models because of the lack of homology between mammalian and *Drosophila* proteins, many chemotherapy drugs can still be tested in *Drosophila* [125,126]. Most of studies about the DNA damage response process of polyploid cells in *Drosophila* focused on the tolerance to γ-ray-induced apoptosis [127].

### 5.1. Radiation-Induced DNA Damage and Apoptosis of Polyploid Cells in Drosophila Models

According to previous studies on *Drosophila*, it has been observed that the polyploid follicle cells in the egg chambers of the endocycling stage (oogenesis stage 7 to 10) do not show apoptosis after γ-irradiation, whereas a significant number of diploid follicle cells in the mitotic stages (stage 1 to 6) exhibit apoptosis [59,60,61]. This observation suggests that polyploid cells possess strong radiation resistance, which is similar to what has been reported in mammalian studies [65]. Indeed, the studies demonstrated that this radiation resistance of polyploid follicle cells is supported by the low level of p53 and silencing of pro-apoptotic genes [59,60,61,62]. The γ-ray-induced apoptosis in the *Drosophila* ovarian follicular epithelia was examined 24 h after irradiation [60]. To further validate these observations, we conducted additional experiments and confirmed that many diploid cells in the mitotic stages show apoptosis but polyploid cells in the endocycling stage do not at the 24 h after irradiation (Figure 4). At the same time, we found that polyploid follicle cells also undergo apoptosis shortly after irradiation (2 to 8 h after irradiation) (Figure 4). We confirmed similar apoptosis patterns in the endocycling polyploid cells induced in wing imaginal discs of *en-Gal4>UAS-fzr*. In the irradiated wing discs, apoptosis was observed in both anterior (diploid) and posterior (polyploid) compartments 2 to 8 h after irradiation, but apoptotic polyploid cells were not observed 24 h after irradiation (Figure 4). The similar pattern of apoptosis phenotypes in both experimental systems (endogenously developed polyploid cells in ovarian follicle cells and artificially induced polyploid cells in wing discs) indicate that polyploid cells possess not only the apoptosis-suppressive mechanisms but also a superior ability of DNA damage response or DNA repair.

### 5.2. DNA Damage and Apoptosis Patterns in Polyploid Cells

Considering that irradiation-induced apoptosis is caused by DNA damage, we examined the pattern of a DNA damage marker, γ-H2Av (H2Av is the *Drosophila* homolog of the mammalian histone H2A variants H2AZ/H2AX), in the polyploid wing disc model (*en-Gal4>UAS-fzr*) and found that the γ-H2Av pattern in polyploid cells is distinct from diploid cells. Under normal conditions without irradiation, nuclei of the polyploid cells displayed a punctate pattern of γ-H2Av signals (Figure 5). Under-replication of DNA where genomic copies are underrepresented in particular regions is commonly observed in endogenously developed polyploid cells of *Drosophila* larval fat body, midgut, salivary gland, and mouse trophoblast giant cells [128,129,130]. Therefore, the Fzr misexpression-mediated endocycling is likely to induce similar under-replication and cause the observed DNA damage. After a short period of irradiation (2 h), those nuclei of the polyploid cells showed an equivalent level of γ-H2Av signals compared to diploid cells, suggesting that polyploid cells do not avoid DNA damage to tolerate irradiation (Figure 5). In both diploid and polyploid cells, the γ-H2Av signal intensity was high in the entire region of the nucleus rather than being punctate patterns. Importantly, despite the increased γ-H2Av signals, the apoptotic rate was much lower in polyploid cells compared to diploid cells. After 24 h of irradiation, diploid cells continued to display strong γ-H2Av signals in the entire region of the nucleus, but the signals in polyploid nuclei reverted to a punctate pattern in which the signal was significantly lower than the nuclei of diploid cells. These observations indicate that polyploid cells may resist irradiation through a more effective or quick DNA damage response (DDR).

### 5.3. DNA Damage Response in Polyploid Tumor Cells

γ-H2Av is the phosphorylated histone 2A variant which is phosphorylated by Ataxia-telangiectasia-mutated (ATM) and Ataxia telangiectasia and Rad3-related (ATR) proteins [131,132,133]. Different types of DNA breaks have different tendencies for ATM and ATR recruitment. The DNA single-strand break (SSB) is sensed by the Replication protein A1 (RPA1) which is necessary to recruit ATR and thus activate the signaling cascade involved in DNA replication and DNA damage response [134]. The sensor for double-strand break (DSB) is a protein complex called MRN complex which is made of meiotic recombination 11 (Mre11), DNA repair protein Rad50, and Nibrin (Nbn) [135].

The study of the Notch activation-induced tumor in *Drosophila* larval salivary glands has shown that genes involved in DNA damage response and repair including *mei-41* (the *Drosophila* homolog of ATR), *RpA-70* (the *Drosophila* homolog of RPA1), *Mre11*, and *Rad50* are significantly upregulated in the polyploid tumors. Furthermore, a long-term observation of the transplanted tumors revealed that the knockdown of each of these genes in the tumors suppresses tumor growth. These tumors contain polyploid giant cells that are not supposed to be observed in the transplanted tumors because the polyploid tumors normally undergo depolyploidization. These data suggest that these DNA-damage response and repair genes are involved in the depolyploidization-mediated tumor progression and that this process shares similar mechanisms with meiosis in which the same genes are involved in ploidy-reducing meiotic division [70].

## 6. Outstanding Questions

### 6.1. How Does Transcriptional Regulation Change in Polyploid Cells?

How do transcription levels and the transcriptome change with changes in ploidy level? How is the chromatin state altered in polyploid cells? We still do not have a clear answer to these basic questions. Several studies of plant cells, however, have shown that polyploidization can lead to differences in chromatin interactions, which may contribute to the expression bias of genes in polyploid cells. For example, a study on soybeans demonstrated that the chromatin looping was different in diploid and polyploid cells, potentially affecting gene expression [136]. Polyploidization can also result in changes in chromatin accessibility, which can influence gene expression. Reduced chromatin accessibility has been observed in the hexaploid chromosome arm of wheat, leading to differences in gene expression levels [137]. These changes in chromatin state can be associated with adaptation to external stress and enhanced tolerance to DNA damage in polyploid cells. This type of research in animal cells, however, has been still limited.

### 6.2. What Limits the Ploidy Level?

In physiological or developmental situations, the ploidy level should be regulated in a genetically programmed manner. For instance, in *Drosophila*, ploidy levels vary from organ to organ: ~16C in ovarian follicle cells; >1024C in nurse cells; ~32C in the subperineurial glia; >1024C in the salivary glands; ~256C in the larval fat body; >128C in the larval midgut [129,138,139]. We can assume that each tissue generates polyploid cells for a specific purpose, and thus there is a particular mechanism that determines the ploidy level in each tissue.

Although the ploidy level is not strictly regulated in the artificially induced polyploid cells or polyploid tumor cells, it seems that there is a constraint on the ploidy level. For instance, in the *Drosophila* wing disc model, the ploidy level of the polyploid cells induced by Fzr misexpression for three days is not significantly higher than the ploidy of two-day misexpression. At the same time, we found that some of those polyploid cells show apoptosis after three days of Fzr expression. In the *Drosophila* salivary gland tumor model, the polyploid tumor cells induced by Notch activation undergo depolyploidization after several rounds of endoreplication. We still do not understand what constraint or mechanism limits the ploidy level in polyploidy-promoting situations.

### 6.3. Why Are Polyploid Cells so Resistant to DNA Damage?

According to the experimental data comparing DNA damage and apoptosis patterns between polyploid and diploid cells suggests that polyploid cells resist irradiation through more effective DNA repair. Polyploid cells are more tolerant to DNA damage but still confront challenges in maintaining genome stability. On the other hand, the polyploid rectal papillar cells in *Drosophila* larvae inactivate several DNA damage responses (DDR) but survive mitosis [140]. During the second larval instar, rectal papillar precursor cells become octaploid through endoreplication and then undergo mitosis. These polyploid papillar cells lack a canonical DDR such as S-phase checkpoints and p53-mediated apoptosis as observed in various cancer cells, but they can still respond to DNA damage with activities of persistent DNA repair signaling and enter mitosis [141]. Although X-ray irradiation does not induce apoptosis in the polyploid papillar cells, these cells show robust γ-H2Av accumulation, as we have observed in the polyploid wing disc cells. In another case, recent study using repeated cytokinesis failure to induce polyploidization of *Drosophila* neural stem cells (neuroblasts) showed that multinucleated polyploid cells can proceed through the cell cycle in an asynchronous manner. In these multinucleated neuroblasts, DNA damage occurs in a subset of nuclei that are not yet competent to enter mitosis when they are exposed to the mitotic environment. Interestingly, the DNA damage generated in a subset of multiple nuclei during mitosis can be attenuated by forcing cell-cycle synchronization [142].

Understanding how the mechanisms of DDR in polyploid cells are different from diploid cells and why cells become so resistant to DNA damage by polyploidization would be very important, especially in the context of cancer therapy to target polyploid cells.

## 7. Concluding Remarks

Malignant tumors that have begun to invade the body have a high recurrence rate even after treatment with surgical resection, anticancer drugs, and radiation therapy. This situation further indicates that the nature of cancer is still not completely understood. Based on the data reported so far, it can be assumed that polyploid cancer cells, which are latent in many malignant tumors, are one of the starting points of cancer recurrence, especially after standard treatment such as radiation therapy. Although the presence of polyploid cells in cancer tissues has been observed and described for more than 150 years, the function and role of these cells in cancer progression in vivo has been studied only in a very small number of cases and remains largely unexplored. For this reason, no therapeutic strategies have been developed that target polyploid cancer cells. This is largely due to the lack of appropriate experimental models to study polyploid cells in vivo. As we have shown in this review, such experimental models are now available in *Drosophila*. In particular, the mechanism of polyploidization via cell cycle transition in development has been well studied in *Drosophila* model systems, such as ovarian follicle cells. Moreover, the recent reports of tumor induction in *Drosophila* epithelial tissues containing polyploid cells, all of which can be easily reproduced in the laboratory using conventional genetic techniques in *Drosophila*, may provide an excellent experimental model for future studies of polyploid cancer cells. With these experimental models in place, we feel that now is the right time to begin research into the nature of polyploid cancer cells that have been elusive for so many years.

## Figures and Tables

**Figure 1 genes-15-00096-f001:**
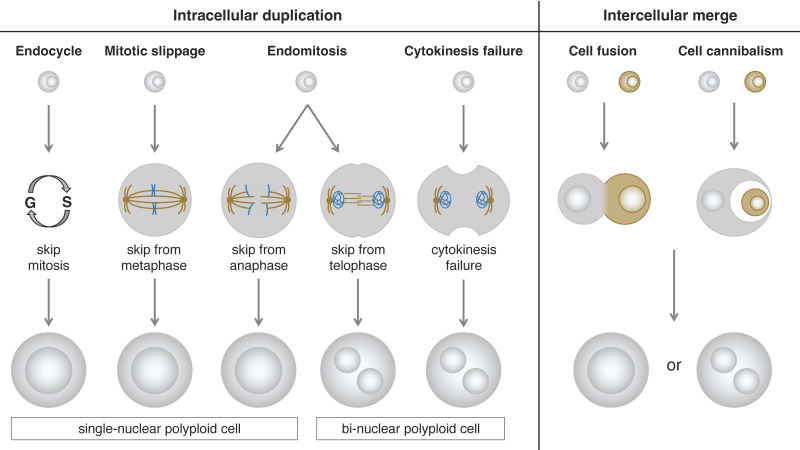
Schematics of the different mechanisms that lead to generation of two types of polyploid cells: The polyploidization processes can be classified into two categories: intracellular duplication and intercellular merge. Based on the cell cycle modifications, the intracellular duplication can be further divided into four different mechanisms: endocycle, mitotic slippage, endomitosis, and cytokinesis failure. The intercellular merge can be also divided into two different processes: cell fusion and cell cannibalism.

**Figure 2 genes-15-00096-f002:**
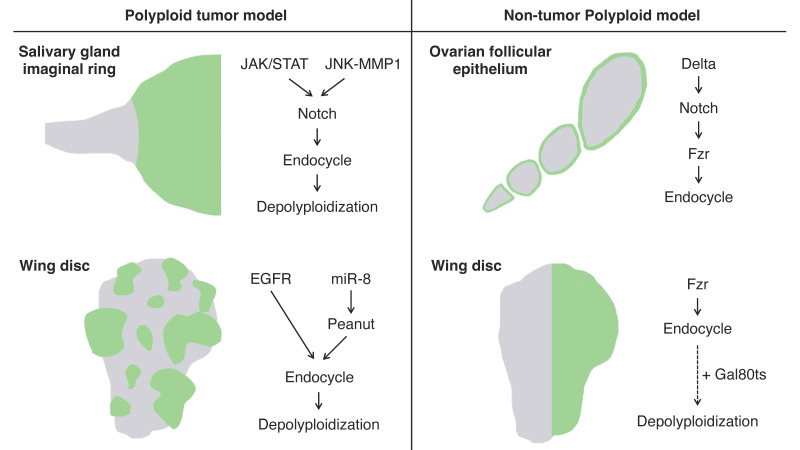
Schematics of the experimental models for polyploidy in *Drosophila*: Left panel: the artificially induced tumor models featuring polyploidization and depolyploidization during the tumor progression induced in the larval salivary glands (**top**) and wing imaginal discs (**bottom**) using the Gal4-UAS system. The green areas indicate polyploid tumors induced by the Gal4-UAS system. Right panel: non-tumor polyploid cell models in the ovarian follicular epithelium (endogenously programmed endocycle) and larval wing imaginal discs (artificially induced endocycle). The green areas indicate the regions where cells are going to become polyploid in these models.

**Figure 3 genes-15-00096-f003:**
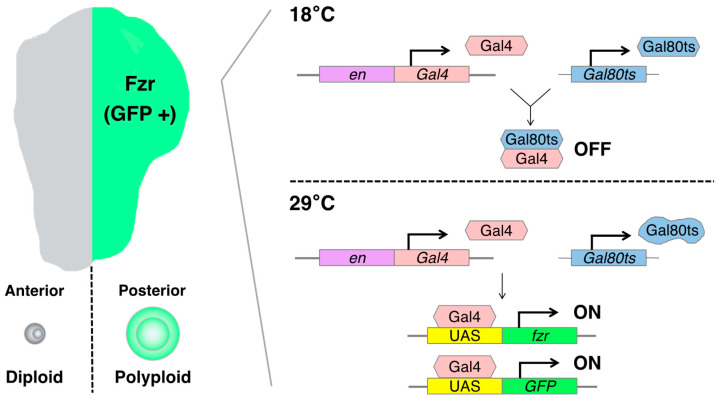
Schematics of the artificially induced polyploidy model using the Gal4-UAS binary expression system in *Drosophila* wing imaginal discs: The *engrailed-Gal4* (*en-Gal4*) transcriptional activator drives expression of the UAS-linked *fizzy-related* (*fzr*) specifically in the posterior compartment of wing imaginal discs, enabling the induction of endoreplication-mediated polyploidization in the posterior half of the tissue. The Gal80 blocks Gal4-mediated transcriptional activation by binding to its transcriptional activation domain. The Gal80ts is functional to inhibit Gal4 activation at 18 °C but ceases to function at 29 °C.

**Figure 4 genes-15-00096-f004:**
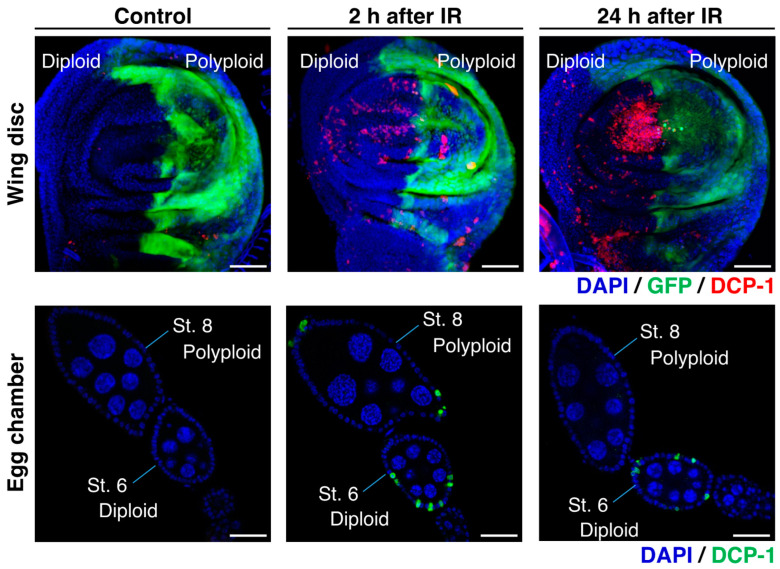
Radiation-induced apoptosis of polyploid cells in *Drosophila* models: Upper panels: Wing imaginal discs with *en-Gal4*-driven *fzr* expression in the posterior compartment (labeled by GFP expression) dissected from third instar larvae after exposure to 0 Gy (**left**) or 35 Gy (**middle** and **right**) of γ-ray irradiation. Lower panels: Wild-type egg chambers dissected from wild-type adult female flies after exposure to 0 Gy (**left**) or 35 Gy (**middle** and **right**) of γ-ray irradiation. The wing discs and the ovaries were stained for DCP-1 (red in the upper panels and green in the lower panels) with an anti-cleaved *Drosophila* Dcp-1 antibody (Cell Signaling Technology). Scale bars represent 50 μm.

**Figure 5 genes-15-00096-f005:**
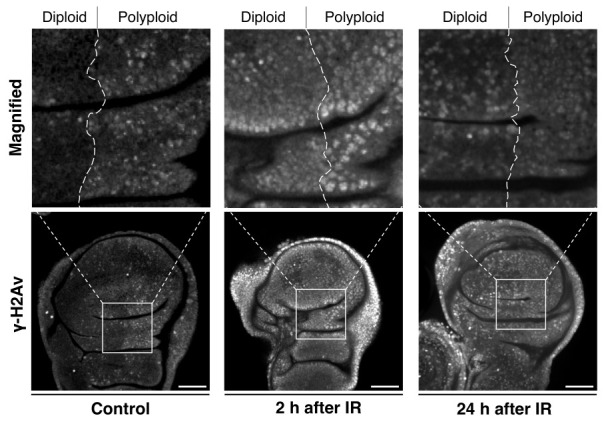
Polyploid cells have an effective DNA damage response: Wing imaginal discs with *en-Gal4*-driven *fzr* expression in the posterior compartment dissected from third instar larvae after exposure to 0 Gy (**left**) or 35 Gy (**middle** and **right**) of γ-ray irradiation. Middle: 2 h after irradiation. Right: 24 h after irradiation. The wing discs were stained for γ-H2Av with anti-γ-H2Av antibody (DSHB). Upper panels: Magnifications of the boxes indicated in the lower panels. Scale bars represent 50 μm.

## Data Availability

Not applicable

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
