# Peer review of "Polyploid Cancer Cell Models in Drosophila"

_genes, 2024, doi:10.3390/genes15010096_

Round 1

Reviewer 1 Report

Comments and Suggestions for Authors

This review adequately describes Drosophila models to study polyploid cells. Unfortunately, the connection between normal polyploid cells in Drosophila and polyploid cancer cells leaves much to be desired. In fact, some of the information is wrong. It is true that cancer cells with multiple centrosomes start mitosis with multipolar spindles, however, multipolar spindles are not observed during anaphase. This is because a strong spindle assembly checkpoint maintains these cells in mitosis for a longer period of time, thereby enabling centrosomes to migrate to two poles. Indeed the study of cell division by video-microscopy has revealed that multipolar division is a rare phenomenon and when it happens it does not produce viable daughter cells because they do not receive the correct set of chromosomes. Hence multipolar division is not the mechanism by which chromosome instability is generated. The vast majority of cancer cells with multiple centrosomes divide by bipolar division. Importantly, the multipolar spindle at the start of mitosis promotes merotelic attachments which lead to a very high frequency of chromosome instability. Notably, the attachment of a chromatid to two centrosomes that go to opposite poles of the cell results in a lagging chromosome. Therefore, most tetraploid cancer cells are not exactly tetraploid, but are sub-tetraploid. Some cells have lost one, two, three or more chromosomes. In each case, different chromosomes are lost. For example, one cell will have lost a copy of chromosomes 1, 13 and 18 whereas the next cell will have lost a copy of chromosomes 3 and 5. This process generates genetic diversity very rapidly within a population. Invariably, whatever selective pressure is applied, a genetic variant will emerge that can survive and thrive in the new conditions. Note also that the tetraploid state also enables the testing of many combinations that are not possible in the context of a diploid cell. Considering the large number of haploinsufficient genes, it is not possible for diploid cells to lose some chromosomes. In contrast, tetraploid cells start with 4 copies of each genes and can then survive the loss of a chromosome copy.

The ability of tetraploid cells to generate immense genetic diversity very rapidly probably explains why they can resist various forms of stress. In this regard, the section “ Polyploidy Promotes Tumorigenesis and Therapeutic Resistance “ fails to provide a mechanistic understanding of therapeutic resistance. The same comment applies to the text on radiation tolerance and stress resistance.

In conclusion, while much information can be gained from the study of specialized normal cells that become polyploid, the parallel with polyploid tumor cells is not necessarily adequate. One must be cautious. The cause and the consequence of polyploidy in cancer cells are quite different.

Reviewer 2 Report

Comments and Suggestions for Authors

The authors discuss the benefits of using Drosophila as a model to study polyploid cancers. The topic is of high interest given the high incidence of polyploidy in cancer and the need for models like Drosophila to better understand the role of polyploidy. Overall the text is understandable to a Drosophila audience or an audience familiar with polyploidy,  but a revised version should make the text more accessible to non-Drosophila and non-polyploid experts.  

General note 1- you often say “our” experimental system. But similar approaches have been used by other labs (Deng, Calvi, Fox, O’Farrell). Please revise. It is fine to say that you have used an approach, but be careful about when to say “our system.”

General note 2- Please qualify discussion of depolyploidization to note that such processes rarely result in a return to a lower euploid state. Rather, such events almost always involve aneuploidy. In at least one case (see below, the authors claim depolyploidization in a case where ploidy is maintained)- please check all other examples in the text carefully

General comment 3-  clarify whether you mean multipolar metaphase only followed by aneuploid bipolar division (as seen by Ganem and Pellman, Nature 2009), or multipolar anaphase and complete multipolar cytokinesis? The latter is rarer in terms of reports in the literature.

Line 33: Please provide a definition of PGCCs. It’s not clear how PGCCs differ from other non-PGCC polyploid cancer cells. At what point do they become “giant?”

Line 60: what is meant by “bunch of proteins”?

Lines 98-119: this paragraph would benefit from a figure diagramming the different roads to polyploidy

Line 555: rectal papillar cells do not undergo depolyploidization but rather maintain ploidy during mitotic divisions

Relevant literature not cited, but should be cited and discussed in the final draft:

Clay et al (2021) JCB- polyploidy favors alternative end joining DNA repair during mitosis in Drosophila rectal papillar cells

Nano et al (2019) Curr Biol- polyploidy and cell cycle asynchrony in the Drosophila brain

O’Neill and Rusan (2022) Development- polyploidy and aneuploidy in the Drosophila mushroom body

Schoenfelder et al (2014) Development- reports complete multipolar division in polyploid Drosophila rectal papillae

Stormo and Fox (2016) eLife and Stormo and Fox (2019) MBoC- induced endocycles in Drosophila wing discs activates the spindle checkpoint and disrupts cohesion organization, leading to aneuploidy

Vidwans et al (2002) Curr Biol- failed sister chromatid separation after induced endocycles in Drosophila embryos

Comments on the Quality of English Language

There are a few spots where English language usage should be corrected. An example of one type of error is found on line 246- "the fzr" should be "fzr" 

Round 2

Reviewer 2 Report

Comments and Suggestions for Authors

the revised manuscript is much improved

Comments on the Quality of English Language

the quality of English language needs to be improved prior to publication. There are numerous grammatical errors and it is very noticeable. 

Author Response

We would like to thank you for your comment. Our manuscript will be checked by the in-house English editor of the Journal before publication.